behaviour, evolution

adverse early life experiences, maternal, paternal, grandparents, matrilateral bias, parental investment

**Author for correspondence:**
Samuli Helle
e-mail: sayrhe@utu.fi

# Matrilateral bias of grandparental investment in grandchildren persists despite the grandchildren's adverse early life experiences

Samuli Helle[1], Antti O. Tanskanen[1,2], David A. Coall[3] and Mirkka Danielsbacka[1,2]

[1]Department of Social Research, Faculty of Social Sciences, University of Turku, Assistentinkatu 7, 20014 Turku, Finland
[2]Population Research Institute, 00101 Helsinki, Finland
[3]School of Medical and Health Sciences, Edith Cowan University, Joondalup WA 6027, Australia

Ⅾ SH, 0000-0002-6216-3759; DAC, 0000-0002-0488-2683

Evolutionary theory predicts a downward flow of investment from older to younger generations, representing individual efforts to maximize inclusive fitness. Maternal grandparents and maternal grandmothers (MGMs) in particular consistently show the highest levels of investment (e.g. time, care and resources) in their grandchildren. Grandparental investment overall may depend on social and environmental conditions that affect the development of children and modify the benefits and costs of investment. Currently, the responses of grandparents to adverse early life experiences (AELEs) in their grandchildren are assessed from a perspective of increased investment to meet increased need. Here, we formulate an alternative prediction that AELEs may be associated with reduced grandparental investment, as they can reduce the reproductive value of the grandchildren. Moreover, we predicted that paternal grandparents react more strongly to AELEs compared to maternal grandparents because maternal kin should expend extra effort to invest in their descendants. Using population-based survey data for English and Welsh adolescents, we found evidence that the investment of maternal grandparents (MGMs in particular) in their grandchildren was unrelated to the grandchildren's AELEs, while paternal grandparents invested less in grandchildren who had experienced more AELEs. These findings seemed robust to measurement errors in AELEs and confounding due to omitted shared causes.

## 1. Introduction

From an evolutionary point of view, grandparental investment (i.e. cost-bearing actions of grandparents that improve the fitness of the recipient) in descending generations results from efforts to maximize their inclusive fitness [1]. Compared to parental investment, the costs of grandparental investment tend to be lower, particularly for post-reproductive adults, and the benefits of grandparental investment in terms of inclusive fitness are predicted to outweigh the costs [2–5]. In recent decades, we have witnessed increased opportunities for grandparental investment due to improved health and increased life expectancy, resulting in longer shared lifespans between grandparents and grandchildren [6,7]. Moreover, because of declining fertility rates in several contemporary high-income post-industrial countries, grandparents currently have fewer grandchildren, meaning that grandparents can potentially invest more in a specific grandchild because of the lower number of alternative investment options [8,9]. Whereas in preindustrial farming and hunter–gatherer societies, where grandparental presence has been found to increase the early life survival rates of grandchildren (e.g. [10–15]), we expect current grandparental

investment to be directed towards skills (e.g. cognitive functioning) and well-being in grandchildren, which determine their success in contemporary high-income post-industrial societies [8,16,17].

An essential part of comprehending evolutionary processes is understanding that adaptive evolution is ultimately a response to changing external abiotic and biotic conditions that affect optimal life-history strategies [18]. For example, cooperative breeding in mammals and birds is regarded as a fitness-enhancing strategy to cope with harsh and unpredictable environments [19,20], and many species show behavioural plasticity to changing environmental conditions [21]. Therefore, within a cooperative breeding context, grandparental investments in subsequent generations are likely to have evolved to track external cues that influence the fitness benefits and costs of such investments.

Previous social scientific literature on grandparental investment has identified several factors, commonly related to the socioeconomic position of the grandchild's family, which may influence grandparental investment (see [22] for a recent review). This literature has also argued that grandparental investment will increase (or compensate) in response to adverse life events in a grandchild's family when the recipient's need for help is higher [23–25]. For example, grandparents have been found to allocate their help to their kin with the greater need [26], and they have been found to be particularly important for grandchildren if there is a severe illness or death of a family member (particularly of the mother), parental divorce, financial hardship in the family, or if the grandchild experiences behavioural problems or harsh parenting [27–32]. Ultimately, in the case of the greatest need when parents are not available, grandparents become the primary caretakers of their grandchildren, providing the largest investment [33].

Previous research has thus investigated grandparental investment in response to an increased need for support by the grandchildren. Here, we propose an alternative scenario based on evolutionary theory that focuses on the balance between the costs and benefits of grandparental investment, resulting from changes in the environments where their grandchildren grow up. It is suggested that increased levels of family instability lead to increased psychological and physical costs (e.g. increased risk of experiencing violence) on the grandparental investment [34], which may decrease investment. In addition, the fitness benefits gained from investing in grandchildren may be reduced, as growing up in adverse environments may reduce the grandchildren's reproductive value, that is, their future prospects for successful survival and reproduction (i.e. their fitness) [35]. Consequently, this may also be selected to decrease grandparental investment. In general, grandparental investment should obey the same logic as parental investment but with differing costs and benefits. Reduced parental investment and an earlier weaning age for children have both been observed in societies living in poor environments suffering from famine, wars and high disease burden [36]. Moreover, socioeconomic deprivation has been associated with lower offspring birth weight [37,38], shorter breastfeeding durations [38,39] and reduced height in adulthood [40]. Such reduced parental investment in children is likely to have long-term consequences for the children's own future parenting behaviour, reproduction, physical and mental health, and

survival (e.g. [41–47]). This likely reduces the reproductive value of the grandchildren and, consequently, the inclusive fitness benefits grandparents receive from investing heavily in them [48]. Finally, recent theoretical modelling of the evolution of early life effects on later performance has suggested that the social environment experienced by individuals in childhood may be more important for their later performance than the abiotic environment experienced [49]. This finding emphasizes the importance of measuring the social aspects of the developmental environment while considering the context specificity of the grandparental investment.

Well-established variations in grandparental lineages and gender mean that not all grandparents invest equally in their grandchildren [22]. Several hypotheses have been suggested to explain the biased pattern of grandparental investment [8]. Paternity uncertainty favours investment by maternal grandmothers (MGMs), followed by maternal grandfathers (MGFs), paternal grandmothers (PGMs) and finally paternal grandfathers (PGFs) [50]. In addition, there is a more general bias in investment towards matrilateral kin due to differential fitness returns through maternal and paternal relatives [51]. Sex-specific reproductive strategies suggest that maternal kin invest more in existing grandchildren whereas paternal kin favour grandchild quantity over quality [52]. In addition, the sex-chromosomal selection hypothesis is based on the differential inheritance of sex chromosomes between genders [53]. These are put forward as potential evolutionary reasons as to why some grandparents invest more than others. Currently, there seems to be no consensus on the relative importance of these hypotheses in explaining the differences in grandparental investment. However, the common finding across a wide range of societies and interdisciplinary research is that MGMs invest the most in their grandchildren and PGFs invest the least [2,3,22,25,54], and that maternal grandparents in general invest more than paternal grandparents [51].

The aim of the current research was to examine how grandparental investment among different grandparent types responds to the adverse early life experiences (AELEs) faced by their grandchildren from birth to adolescence. Specifically, our goal was to contrast the expectations from the social scientific literature suggesting that grandparents should increase their investment in grandchildren owing to increases in need [22] with the expectation from evolutionary theory, suggesting that when the expected fitness returns decrease (and/or the costs increase), grandparental investment will diminish. Moreover, we expect different responses between different grandparent types: when compared to maternal grandparents (and MGMs in particular), the investment of paternal grandparents should be more facultative [22,55,56]. Since prior research has shown that maternal grandparents are willing to expend extra effort to invest in their grandchildren when compared to other grandparent types (e.g. [55]), we expect the investment of maternal grandparents and MGMs in particular to be the least sensitive, while the investment of paternal grandparents and PGFs in particular to be the most sensitive to changes in their grandchildren's early developmental environment. Our goal was not to contrast the differences between all grandparental types but to concentrate on two pre-defined hypotheses:

> Hypothesis 1: The more AELEs a grandchild has, the more the investment from the paternal grandparents will change when compared to that of the maternal grandparents.

Hypothesis 2: As a grandchild's AELEs increase, the investment of the maternal grandmother will change the least when compared to the other grandparent types.

To examine these hypotheses, we used nationally representative population-based data gathered from English and Welsh adolescents (ranging in age from 11 to 16) and a structural equation modelling (SEM) framework [57]. SEM has the benefit of allowing the fitting of complex structural models, the incorporation of measurement error into the constructs of scientific interest using unobserved latent variables and enabling various sensitivity checks for the robustness of the results to different modelling assumptions (e.g. to external model misspecification).

## 2. Methods and materials

### (a) Data

We used the data from the Involved Grandparenting and Child Well-Being 2007 survey collected by the GfK National Opinion Polls, which is a nationally representative sample of English and Welsh adolescents aged 11–16 years [28,58,59]. In every selected school, classes were randomly chosen. Larger schools had a greater probability of being included in the final sample, and the response rate was 68%. Respondents completed the questionnaire in a school classroom, and the original sample included 1566 adolescents [28,60]. When filling in the questionnaire on grandparental investment, respondents were asked to answer questions relating to only those grandparents who were still alive, i.e. only those with at least one living grandparent were considered ($n = 1488$). As is common practice in previous research, we also excluded those children from the analyses who resided with their grandparents ($n = 58$). This was because, based on the data, we could not separate the cases where grandparents were the sole caretakers of the grandchildren (in which case their investment is much more obligatory) from those cases of three-generation households. The total number of children included in the analysis was 1430. For descriptive statistics, please refer to table 1.

### (i) Measurement of a grandchild's adverse early life experiences

To measure the grandchildren's AELEs, the distal adverse life events scale of Tiet *et al.* [61] was used to record the total number of adverse life events experienced throughout the child's life up to until 1 year before the response was collected. In its original formulation, the Adverse Life Event scale consists of 25 possible events that children have little or no control over. However, the original scale included several events that were not severe enough to have a meaningful influence on grandparental investment in the context of current research. Therefore, only the following events (answered by yes/no) that directly involved the grandchild or her/his family were used to calculate the number of AELEs for each grandchild in earlier life: 'someone in the family died', 'there was a negative change in parent's financial situation', 'family had drug/alcohol problem', 'respondent got seriously sick or injured', 'respondent was a victim of crime/violence/assault', 'parents separated or divorced' and 'one of the parents went to jail'. In addition, we also included the question 'have you ever qualified for free school meals (even if not taken)?'. As children from

low-income families receive free school meals in the UK, this variable indicates the financial conditions of the family. Thus, the resultant composite index for AELEs was the sum of eight indicators.

### (ii) Grandparental investment

To measure grandparental investment in their grandchildren, we used questions developed by Elder & Conger [62]. From the list of all questions available, we chose four questions that directly measured grandparental investment; these were 'how often do you see them' (Q15), 'their grandparents had looked after them' (Q26), 'they could depend on their grandparents' (Q27) and 'provided financial assistance or help' (Q38). Question Q26 was reverse-scaled to match the meaning and ordering of the other scales. Questions Q15, Q26 and Q27 were measured on a 4-point Likert-type scale ranging from 1 = *not at all/never* to 4 = *a lot/every day*, and Q38 was measured on a 3-point Likert-type scale ranging from 1 = *never* to 3 = *usually*.

### (b) Statistical analysis

We used SEM with multiple-indicator latent variables [57] to examine how the grandchildren's AELEs influenced grandparental investment. The benefits of using SEM with latent variables instead of the regular observed variable regression models include the potential for more generalizable results owing to a more causally oriented approach, handling of measurement error in the latent constructs of interest that produce more consistent regression estimates and increased statistical power, and more flexible representations of the data with the additional parameters [63].

The response variable used was a latent variable measuring 'grandparental investment' for each grandchild with four effect indicators (i.e. the latent variable causes variation in its indicators) that were the questions asked from the grandchildren [57]. The question 'provided financial assistance or help' was regarded as the most relevant measured variable of investment and was thus used as a marker indicator of the latent variable by fixing its unstandardized loading to unity. This sets the scale for the latent variable, which also has an error term (or disturbance), representing the fact that not all of its causes are modelled here [57]. In other words, the use of latent variables allows for the inclusion of measurement errors when measuring scientific constructs. The effect indicators for this latent variable were treated as ordinal variables that were modelled using the probit link function. Hence, these loadings can be interpreted as the extent to which a one-unit increase in the latent variable score, defined by its marker indicator, changes the predicted probit index in standard deviation units of the latent response variable that is connected to its observed indicators by their threshold structure (for more details, please see electronic supplementary material, figure S1) [57]. To compare how grandparental investment among the grandparent types varied with the AELEs, we needed to first establish measurement invariance (i.e. homogeneous measurement properties) for grandparental investment between the grandparent types [64]. Analysis of measurement invariance showed that we had to rely on partial measurement invariance, where one of the four-factor loadings was non-invariant between the groups (see electronic supplementary material, table S1). Thus, we

**Table 1.** Descriptive statistics for the variables used in this study. MGM, MGF, PGM and PGF stand for maternal grandmother, maternal grandfather, paternal grandmother and paternal grandfather, respectively.

|  | mean | s.d. | min. | max. |
|---|---|---|---|---|
| *characteristics of the grandchild* | | | | |
| grandchild age (mean) | 13.4 | 1.4 | | |
| don't know/missing (%) | 0.56 | | | |
| ethnicity (%) | | | | |
| white | 89.0 | | | |
| Black or Afro-Caribbean | 3.5 | | | |
| Asian | 2.1 | | | |
| mixed | 2.2 | | | |
| don't know/missing | 3.2 | | | |
| AELEs | 1.6 | 1.4 | 0 | 8 |

|  | MGM | MGF | PGM | PGF |
|---|---|---|---|---|
| *characteristics of the grandparent* | | | | |
| number of other grandchildren (%) | | | | |
| respondent is the only grandchild | 2.6 | 1.7 | 2.0 | 1.8 |
| 2 or 3 | 22.7 | 18.6 | 18.3 | 13.5 |
| >3 | 53.4 | 40.9 | 44.6 | 33.6 |
| don't know/missing | 21.2 | 38.8 | 35.2 | 51.1 |
| living distance between the grandparent and the grandchild (%) | | | | |
| overseas | 7.1 | 5.1 | 5.2 | 4.1 |
| further away in the UK | 18.8 | 16.4 | 20.0 | 15.5 |
| not in the same town but within 10 miles | 20.6 | 15.0 | 18.0 | 14.5 |
| in the same town | 34.5 | 26.4 | 24.1 | 16.6 |
| don't know/missing | 19.1 | 37.1 | 32.7 | 49.3 |
| grandparent age (%) | | | | |
| < 50 years | 0.9 | 0.6 | 0.4 | 0.4 |
| 50–60 years | 12.0 | 6.2 | 5.4 | 2.9 |
| 60–70 years | 34.5 | 26.4 | 28.3 | 20.6 |
| > 70 years | 24.6 | 22.0 | 22.9 | 19.6 |
| don't know/missing | 28.0 | 45.0 | 43.0 | 56.6 |
| *variables measuring grandparental investment* | | | | |
| 'Do they give you money or help in any other way?' (%) | | | | |
| never | 7.7 | 9.7 | 10.3 | 10.1 |
| occasionally | 28.8 | 21.9 | 26.0 | 20.0 |
| usually | 45.5 | 33.9 | 33.7 | 24.3 |
| missing | 18.0 | 34.6 | 30.0 | 45.7 |
| 'How often do you see them?' (%) | | | | |
| never | 4.9 | 5.5 | 5.9 | 6.6 |
| several times a year | 30.8 | 25.4 | 33.5 | 25.5 |
| twice a week | 30.2 | 22.8 | 23.4 | 17.6 |
| daily | 16.4 | 10.7 | 6.4 | 3.7 |
| missing | 17.7 | 35.6 | 30.8 | 46.6 |
| 'How often do your grandparents look after you?' (%) | | | | |
| every day | 4.0 | 2.7 | 1.0 | 0.6 |
| once a week or so | 20.8 | 15.2 | 12.8 | 9.3 |
| several times a year | 32.9 | 23.9 | 26.4 | 18.6 |

(*Continued.*)

**Table 1.** (Continued.)

| | MGM | MGF | PGM | PGF |
|---|---|---|---|---|
| never | 24.3 | 24.1 | 30.0 | 26.4 |
| missing | 17.9 | 34.1 | 29.8 | 45.1 |
| 'How much can you depend on your grandparent to be there when you really need him/her?' (%) | | | | |
| not at all | 8.9 | 9.7 | 12.1 | 11.5 |
| a little | 11.8 | 10.8 | 12.4 | 9.6 |
| sometimes | 18.5 | 14.7 | 17.7 | 13.7 |
| a lot | 43.2 | 30.6 | 27.8 | 19.8 |
| missing | 17.8 | 34.2 | 30.0 | 45.4 |

proceeded with a multi-group SEM by simultaneously fitting our model to all four grandparent types.

The main predictor variable that influenced the latent 'grandparental investment' for the grandchild was the number of AELEs. Furthermore, as precision covariates aimed to reduce the error variance in 'grandparental investment' (i.e. not causal confounders of the associations studied here; see e.g. [65]), the model fitted included a living distance between the grandparent and the grandchild (in the same town, not in the same town but within 10 miles, further away in the UK, or overseas (=a reference category)), the number of other grandchildren (single grandchild (=a reference category), one to two grandchildren, more than three grandchildren, two to four grandchildren, more than four grandchildren and more than six grandchildren), grandchild's ethnicity (white (=a reference category), black or Afro-Caribbean, Asian, and mixed parentage), sex (female (=a reference), male) and grandparental age (less than 50 (=a reference category), 50–60, 60–70 and more than 70 years of age). As a potential statistical confounder, we included grandchild age into the analysis as grandchild's age is likely linked to the probability of being exposed to a number of adverse early life events (younger children may not yet have experienced so many adverse events as older children) and effects of grandparental investment vary by grandchild age, with grandparents investing more in young grandchildren. All the categorical variables were dummy coded, and grandchild age was grand mean centred. All the regression parameters as well as the latent variable intercepts (except for MGMs, which was fixed to zero for identification purposes) and variances were allowed to vary between the grandparent types. As indicated by the measurement invariance analysis (see electronic supplementary material), a single residual covariance among the factor indicators was also included in the model.

To handle missing data in the independent variables, we used multiple imputation and followed the guidelines given by von Hippel [66] for the number of imputed datasets needed. By accepting a 5% change in the standard error of the estimates, we imputed 62 datasets using a multilevel variance covariance approach (as we had missing data on variables measured within and between grandchildren) using an unrestricted model with a Bayesian estimator [67]. Because of the multiple imputations, we used Wald's test instead of the likelihood ratio test. Since the two pre-planned tests were not orthogonal, we controlled for the type I error rate by using the false discovery rate (FDR) among the set of rejected hypotheses [68]. FDR, defined as the expected proportion of type I errors among all significant results, corrected for the expected proportion of type I errors by providing adjusted $p$-values. The mean- and variance-adjusted diagonally weighted least-squares (WLSMV) estimator and default delta parametrization were used for the estimation of multi-group SEM. As we had multiple observations per grandchild (i.e. a maximum of four living grandparents), the grandchild's identity was used as a design-based clustering factor to obtain unbiased estimates and robust standard errors [69].

We performed two robustness checks for the model assumptions that could have affected our results. First, using a single-indicator latent variable approach to account for measurement error when measuring AELE, we examined the sensitivity of our results for the artificially reduced reliability (down to 60% reliability) of the AELE variable [70]. It is well known that measurement errors in predictors attenuate their regression coefficients toward zero (e.g. [71]). Such lowered reliability (i.e. increased measurement error) might include the factors missed by the original scale to represent the grandchild's AELEs and, for example, errors in the participants' understanding of the questions asked. In this approach, the original observed predictor (i.e. AELEs) is replaced by a latent variable for which measured AELEs has a unit loading, while its error variance is set by multiplying its sample variance with the desired level of reliability. Second, to evaluate the importance of unobserved confounding (i.e. shared causes for both independent and dependent variables) on the association between grandparental investment and the AELEs of grandchildren, we applied the method recently described by Harring et al. [72]. In this method, the effect of a potential unmeasured confounder(s) is mimicked by a phantom variable that affects both the predictor (i.e. AELEs) and the outcome (i.e. grandparental investment) in the model. Phantom variables are latent variables without any indicators, precluding the need for actual data. Instead, the mean and variance of the phantom variables are fixed constants, usually set to zero and unity, respectively [72]. The rationale is to examine the sensitivity of the original conclusions when one adds the phantom variable as unmeasured confounder(s) into the model and varies the strength of the expected confounding. One potential confounder for the true causal effect between AELEs and grandparental investment that is unmeasured in these data is the socioeconomic status of the grandparents, acting via or parallel to their children's (i.e. the parents of the

grandchildren) socioeconomic status. It is likely that a high socioeconomic status in grandparents acts to reduce the grandchildren's AELEs and increases their investment in their grandchildren. Thus, we concentrated on this scenario. It is of note that depending on the application, the confounder could also be expected to exert positive or negative effects on both the independent and dependent variables or positive effects on the predictor and negative effects on the outcome. While the signs of the suspected confounder effects are usually easy to imagine, the strength of these effects is usually arbitrary without strong prior knowledge. Hence, using a range of values is recommended [72]. Here, we started the process of fitting different values to the strength of the confounding value based on the observed associations between the AELEs and investment among the grandparents and by gradually increasing the strength of the confounding variable until the confounding changed the statistical inference of the association of interest. Mplus 8.5 [73] was used for all data analyses.

## 3. Results

The multi-group SEM showed an almost statistically significant difference (when considering the conventional alpha level of 5%) in the regression coefficients for AELEs and grandparental investment between the maternal and paternal grandparents (Hypothesis 1: maternal grandparents: $\beta$ (95% confidence interval [CI] $-0.012$ [$-0.032$, $0.008$], paternal grandparents $\beta$ [95% CIs] $-0.040$ [$-0.064$, $0.017$], $\chi_1^2 = 3.371$, $p = 0.066$, $p_{\text{FDR-adjusted}} = 0.066$). Likewise, the investment of the MGMs seemed less sensitive to increases in the AELEs when compared to the other grandparents (Hypothesis 2: MGMs: $\beta$ (95% CIs) $-0.003$ ($-0.028$, $0.021$), other grandparents: $\beta$ (95% CIs) $-0.036$ ($-0.055$, $-0.016$), $\chi_1^2 = 4.182$, $p = 0.041$, $p_{\text{FDR-adjusted}} = 0.066$). These effect sizes can be evaluated in terms of their predicted marginal effects for the question setting scale of the latent variable 'investment' (i.e. 'provided financial assistance or help'). For example, grandchildren who experienced none of the AELEs compared to those who experienced all eight AELEs had conditional probabilities of scoring 'usually' (i.e. the highest category) of 53.4% and 49.6%, respectively, for the maternal grandparents. For the paternal grandparents, the percentages for grandchildren with zero and eight AELEs were 53.4% and 40.7%, respectively. The corresponding numbers for the contrast between MGMs and other grandparents were 53.4% and 52.4% for MGMs and 53.4% and 42.0% for all other grandparents.

When allowing for differing associations between AELEs and grandparental investment for each grandparent type, we see a clear gradient in these associations across the different grandparent types (table 2; see electronic supplementary material, table S2 for full results). That is, the association between AELEs and grandparental investment was the weakest among MGMs and the strongest negative association was among PGFs. Associations for the MGFs and PGMs fell in between.

Robustness checks showed that reducing the reliability of a variable measuring AELEs down to 60% did not have any effect on the results (table 3). The impact of adding a confounder to the model that negatively affected the number of AELEs but positively affected grandparental investment on grandchildren by varying degrees is given in table 4. A confounder

**Table 2.** Selected results from a multi-group SEM examining how AELEs of grandchildren influenced grandparental investment among MGMs, MGFs, PGMs and PGFs. For the full results, please see electronic supplementary material, table S2. 95% CI denotes 95% confidence intervals of the regression coefficients.

| | $\beta$ | 95% CI |
|---|---|---|
| regression coefficient | | |
| MGMs | $-0.003$ | $-0.028$, $0.021$ |
| MGFs | $-0.025$ | $-0.057$, $0.007$ |
| PGMs | $-0.032$ | $-0.061$, $-0.002$ |
| PGFs | $-0.051$ | $-0.088$, $-0.015$ |

whose effect on both grandparental investment and AELEs was 0.25 and $-0.25$ units, respectively, was strong enough to change our statistical inference on the non-existing association between AELEs and grandparental investment among MGMs ($\beta$ (95% CIs) $-0.003$ ($-0.028$, $0.021$)) to a significant *positive* association of AELEs for investment on grandchildren ($\beta$ (95% CIs) $0.029$ ($0.005$, $0.053$)). In other words, the effect of confounding would have to be approximately 83-times stronger compared to the estimated association between AELEs and the MGMs's investment in order to change our conclusion regarding the parameter. The corresponding values for changing our statistical inference for the association between AELEs and grandparental investment for MGFs, PGMs and PGFs resulted in roughly 14-, 3-, and 9-times stronger confounder effects, respectively (table 4).

## 4. Discussion

The aim of this investigation was to challenge the currently held expectation in social scientific literature that grandparents will increase their investment in grandchildren if the children have an increased need [22], with an opposite prediction that arises from evolutionary theory. Although our results were on the borderline of conventional statistical significance after adjusting for multiple testing, they do suggest that grandparents of grandchildren who experienced more AELEs may provide less grandparental investment. Moreover, the investments made by maternal grandparents, and MGMs in particular, were insensitive to the AELEs experienced by the children. Paternal grandparents, however, invested less in grandchildren who had experienced more AELEs. These results were robust to potential measurement errors in our measure of AELEs, and the confounding (i.e. bias due to omitted shared causes) would have to be much stronger than the associations found here between the AELEs and grandparental investment to alter our conclusions.

Thus, our results support evolutionary prediction, which states that the investments made by grandparents should relate to their potential costs and benefits, as these eventually could determine the payoff in terms of their grandchildren's evolutionary fitness (i.e. their reproductive value measuring the fraction of a future population that has descended from them) [35,74]. In other words, high investment is likely to have poor inclusive fitness payoffs when made in a grandchild who has low reproductive value and is thus selected

**Table 3.** Regression coefficients and their 95% confidence intervals (CI), assessing how AELEs influenced grandparental investment in relation to declining reliability, R, of the variable measuring AELEs. MGMs, maternal grandmothers; MGFs, maternal grandfathers; PGMs, paternal grandmothers; PGFs, paternal grandfathers.

| R | MGMs β (95% CI) | MGFs β (95% CI) | PGMs β (95% CI) | PGFs β (95% CI) |
|---|---|---|---|---|
| 1 | −0.003 (−0.028, 0.021) | −0.025 (−0.057, 0.007) | −0.032 (−0.061, −0.002) | −0.051 (−0.088, −0.015) |
| 0.95 | −0.004 (−0.029, 0.147) | −0.026 (−0.059, 0.006) | −0.035 (−0.065, −0.004) | −0.056 (−0.092, −0.020) |
| 0.90 | −0.004 (−0.030, 0.022) | −0.028 (−0.062, 0.006) | −0.037 (−0.069, −0.005) | −0.059 (−0.097, −0.021) |
| 0.85 | −0.004 (−0.032, 0.024) | −0.030 (−0.066, 0.007) | −0.039 (−0.073, −0.005) | −0.063 (−0.103, −0.022) |
| 0.80 | −0.004 (−0.034, 0.025) | −0.031 (−0.070, 0.007) | −0.041 (−0.078, −0.005) | −0.066 (−0.110, −0.023) |
| 0.75 | −0.005 (−0.036, 0.027) | −0.033 (−0.074, 0.008) | −0.044 (−0.083, −0.005) | −0.071 (−0.117, −0.025) |
| 0.70 | −0.005 (−0.039, 0.029) | −0.036 (−0.080, 0.008) | −0.047 (−0.089, −0.006) | −0.076 (−0.126, −0.026) |
| 0.65 | −0.005 (−0.042, 0.031) | −0.038 (−0.086, 0.009) | −0.051 (−0.096, −0.006) | −0.082 (−0.136, −0.028) |
| 0.60 | −0.006 (−0.046, 0.034) | −0.042 (−0.093, 0.010) | −0.055 (−0.104, −0.007) | −0.089 (−0.147, −0.030) |

**Table 4.** Regression coefficients and their 95% confidence intervals (CI) for AELEs that influenced grandparental investment in relation to varying strengths of unmeasured confounding. MGMs, maternal grandmothers; MGFs, maternal grandfathers; PGMs, paternal grandmothers; PGFs, paternal grandfathers. The italicized coefficients indicate the level of confounding that changed the statistical inference of the association between AELEs and grandparental investment.

| confounding (investment, AELEs) | MGMs β (95% CI) | MGFs β (95% CI) | PGMs β (95% CI) | PGFs β (95% CI) |
|---|---|---|---|---|
| (0, 0) | −0.003 (−0.028, 0.021) | −0.025 (−0.057, 0.007) | −0.032 (−0.061, −0.002) | −0.051 (−0.088, −0.015) |
| (0.001, −0.001) | −0.004 (−0.027, 0.020) | −0.025 (−0.056, 0.006) | −0.033 (−0.062, −0.004) | −0.053 (−0.088, −0.019) |
| (0.01, −0.01) | −0.003 (−0.027, 0.020) | −0.025 (−0.056, 0.006) | −0.033 (−0.062, −0.004) | −0.053 (−0.088, −0.019) |
| (0.05, −0.05) | −0.002 (−0.026, 0.021) | −0.024 (−0.055, 0.007) | −0.032 (−0.061, −0.003) | −0.052 (−0.086, −0.017) |
| (0.1, −0.1) | 0.002 (−0.022, 0.025) | −0.02 (−0.051, 0.011) | *−0.028 (−0.057, 0.001)* | −0.048 (−0.083, −0.013) |
| (0.2, −0.2) | 0.017 (−0.006, 0.041) | −0.005 (−0.036, 0.026) | −0.013 (−0.042, 0.017) | −0.033 (−0.067, 0.002) |
| (0.25, −0.25) | *0.029 (0.005, 0.053)* | 0.007 (−0.024, 0.038) | −0.001 (−0.03, 0.028) | −0.021 (−0.056, 0.013) |
| (0.3, −0.3) | 0.043 (0.019, 0.067) | 0.021 (−0.01, 0.052) | 0.013 (−0.016, 0.043) | −0.007 (−0.042, 0.028) |
| (0.35, −0.35) | 0.060 (0.036, 0.084) | *0.037 (0.006, 0.064)* | 0.030 (0.001, 0.060) | 0.010 (−0.025, 0.045) |
| (0.4, −0.4) | 0.079 (0.055, 0.104) | 0.057 (0.025, 0.088) | 0.049 (0.020, 0.079) | 0.029 (−0.006, 0.064) |
| (0.5, −0.5) | 0.101 (0.076, 0.126) | 0.078 (0.046, 0.110) | 0.071 (0.041, 0.102) | *0.051 (0.015, 0.087)* |

against by natural selection. Given the wealth of evidence on how early life and childhood experiences affect adult phenotypes in humans (e.g. [41–47]), there is probably a link between early environmental and social conditions and an individual's reproductive value in our species. A common expectation among scholars examining these questions is that accelerated reproductive timing commonly related to AELEs or environmental conditions is adaptive by increasing their relative lifetime fitness (e.g. [44]), thus contrasting our rationale for the reduced reproductive value of individuals developing in adverse conditions. However, to the best of our knowledge, no previous studies have established such a link in humans, nor do theoretical life-history models support such conclusions [75,76]. A recent long-term study in wild baboons failed to find to support for accelerated reproductive scheduling in response to early life adversity being adaptive in this species [77]. More studies are needed to link early life conditions with multigenerational investments in humans. This research would benefit from advanced demographic modelling, as done in the case of sex allocation [78], by

making explicit predictions as to how grandparental investment should be expected to vary according to the reproductive value of grandchildren.

The current findings also strengthen our knowledge regarding the important role played by maternal grandparents, and MGMs in particular, as investors in grandchildren [3,54]. The novel contribution of the current research is that for grandchildren who experienced many AELEs the importance of maternal grandparents, and especially MGMs, still seems to hold. Those grandparents who are predicted to invest most in their grandchildren (i.e. maternal grandparents and MGMs in particular) seem to do so irrespective of the cues indicating declining reproductive values in the grandchildren. It is known that supportive caregiving can protect children from for example early adversity-induced psychopathology [79]. MGMs may thus provide such support for their grandchildren, particularly if the care from their primary caretakers is compromised. Also in killer whales, post-reproductive females are found to be particularly important for their relatives in the years when food (salmon)

abundance is low [80]. By contrast, paternal grandparents seem much more sensitive to investing in grandchildren with reduced fitness payoffs, as their investments decreased. From an evolutionary perspective, this may indicate that (i) maternal grandparents, especially the MGMs, can better tolerate for the decreased fitness payoffs in their grandchildren (e.g. owing to known relatedness) or (ii) that the costs of AELEs in grandchildren and in their family are disproportionally taking a toll on paternal grandparents.

The data used in this study have several strengths. The adolescents provided information on grandparental investment and background variables related to themselves, their families and grandparents. Grandparents may not be the ideal source of such information, because since the norm in contemporary high-income post-industrial societies is to treat all children equally, they may try to present their investment as equal in all grandchildren [8,22]. Parents, in turn, may think of grandparents as couples, meaning they may not accurately report the amount of grandparental investment within lineages. Moreover, if one is interested in the investment of all four grandparent types, it would be very complicated to ask either grandparents or parents about the grandparental investment according to all the different grandparent–grandchild dyads. Due to the limitations related to surveying parents and grandparents, children could be regarded as the most reliable source of information on biased grandparental investment [52]. Finally, the conclusions drawn from these data are further strengthened by the modelling framework applied, which enables the evaluation of the two important threats to statistical inference from correlative data: measurement error in predictors and confounding by omitted shared causes [81,82]. Neither lowering the reliability of AELEs down to 60% nor modelling very strong confounding effects compared to the estimated association between AELEs and grandparental investment by grandparent type seemed to change the statistical inference.

The limitations of this research are related to the fact that we could not separate the factors affecting the need for investment from those affecting the cost and benefits of investment using the current data. These aspects are not necessarily synonymous, as not all factors that increase the need for grandparental investment necessarily imply increased costs and/or benefits. Moreover, it has recently been acknowledged that it is generally difficult to quantify and measure environmental unpredictability and harshness that could be relevant for individual development and its long-term consequences [83,84]. We aimed to alleviate such concerns by using a cumulative measure of AELEs instead of relying on only a single or few events that may poorly capture the range of all external cues relevant for parental and grandparental investment [85,86] and using a questionnaire that captures the AELEs during the grandchildren's lifetime. Furthermore, by modelling measurement error in our measure of

AELEs, we showed that our conclusions were robust to such a threat. However, we do not claim to have revealed causal effects here: all the effects modelled were linear and additive within grandparents whereas causal inference is inherently non-parametric in nature [81]. In addition, although the AELEs measured presumably took place before the children were asked to evaluate their grandparents' investment, we cannot exclude the possibility that grandparental investment may have played some role in the level of adversity experienced by the grandchildren at some point in their life. Longitudinal data on adverse events and subsequent changes in grandparental investment are needed to shed more light on potential causal interpretation of our findings.

In conclusion, our results suggest that when considering the AELEs of grandchildren using a cumulative lifetime measure of adverse events instead of a single event or few events, there is more convincing evidence for a decrease in grandparental investment than an increase. An increase in grandparental investment with the increasing need on the grandchild's behalf has been commonly argued in social scientific literature. Here, we provide an alternative evolutionary rationale based on the reproductive value (i.e. fitness) of grandchildren, as we may expect a decrease in grandparental investment if grandchildren experience several AELEs. The between-grandparent patterns found here are also in line with the existing literature, showing the importance of MGMs as the main investors in grandchildren, even in cases where the grandchildren have already faced many hardships in their lives.

**Ethics.** The data used in the current study were approved and the research was performed in accordance with the guidelines of University of Oxford Research Committee. All the participants and their parents gave written consent to participate in the study in accordance with the Declaration of Helsinki.

**Data accessibility.** The data we used in this study are freely available from https://beta.ukdataservice.ac.uk/datacatalogue/studies/study?id=6075#!/details, but please be aware that, as the data are 'safeguarded' (https://www.ukdataservice.ac.uk/get-data/data-access-policy), a user will be required to register for the UK Data Service to be able to access the data. The authors did not have any special access privileges to these data that future researchers would not have. The data are provided in the electronic supplementary material [87].

**Authors' contributions.** S.H.: data curation, formal analysis, methodology, visualization, writing—original draft and writing—review and editing; A.O.T.: conceptualization, funding acquisition, project administration and writing—review and editing; D.A.C.: resources and writing—review and editing; M.D.: conceptualization, funding acquisition, project administration and writing—review and editing.

All authors gave final approval for publication and agreed to be held accountable for the work performed therein.

**Competing interests.** We declare we have no competing interests.

**Funding.** This work was supported by the Academy of Finland (grant nos 317808, 320162, 325857 and 331400) and the Strategic Research Council (grant no 345183).

**Acknowledgements.** We are grateful to Ann Buchanan and the research team for making the Involved Grandparenting and Child Well-Being survey data available via UK Data Service.

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
