## [Peer Review File · Proceedings of the Royal Society B: Biological Sciences]

Review History

RSPB-2021-1798.R0 (Original submission)

Review form: Reviewer 1 (Paula Sheppard)

Recommendation

Accept with minor revision (please list in comments)

Scientific importance: Is the manuscript an original and important contribution to its field?

Excellent

General interest: Is the paper of sufficient general interest?

Excellent

Quality of the paper: Is the overall quality of the paper suitable?

Excellent

Is the length of the paper justified?

Yes

Should the paper be seen by a specialist statistical reviewer?

No

Do you have any concerns about statistical analyses in this paper? If so, please specify them explicitly in your report.

No

It is a condition of publication that authors make their supporting data, code and materials available - either as supplementary material or hosted in an external repository. Please rate, if applicable, the supporting data on the following criteria.

Is it accessible?

Yes

Is it clear?

Yes

Is it adequate?

Yes

Do you have any ethical concerns with this paper?

No

Comments to the Author

Matrilateral bias of grandparental investment in grandchildren persists despite the grandchildren's adverse early life experiences.

This paper examines if grandparents invest more in grandchildren who have had adverse childhood experiences. They find that maternal grandmothers invest highly regardless of the adversity of the childhood while paternal grandparents reduce investment in adversely affected children. The authors use thoughtful statistical methods to get at the proposed causal effects. I was not familiar with the phantom confounder method but at face value it seems a sensible approach overall. It is well written and clear and I think this paper is good and a good fit for PRSB.

I have a few comments below:

Lines 49, 55, 387: Please replace 'Western' and 'modern-day' society with terms that depict the relevance of the context more precisely (e.g. low fertility, high income, post-industrial). Western has political connotations and modern-day is derogatory to lower income countries which are also 'modern-day' insofar as they are contemporary.

Lines 66-69 talk about the literature on grandparents influence on grandchildren's socioeconomic position as well as whether this is a function of grandchild needs. I was surprised that the recent review by Anderson et al 2018 was not referred to as it addresses both these topics extensively.

P11: Please provide proper justification for the control variables chosen. Most of them make sense to me (although not grandchild ethnicity or sex), nevertheless justification for each one should be clearly stated in the methods section.

Anderson, L, et al. 2018. 'Grandparent Effects in Educational Outcomes: A Systematic Review'. Sociological Science 5: 114-42.

Review form: Reviewer 2

Recommendation

Reject - article is not of sufficient interest (we will consider a transfer to another journal)

Scientific importance: Is the manuscript an original and important contribution to its field?

Acceptable

General interest: Is the paper of sufficient general interest?

Marginal

Quality of the paper: Is the overall quality of the paper suitable?

Marginal

Is the length of the paper justified?

Yes

Should the paper be seen by a specialist statistical reviewer?

Yes

Do you have any concerns about statistical analyses in this paper? If so, please specify them explicitly in your report.

Yes

It is a condition of publication that authors make their supporting data, code and materials available - either as supplementary material or hosted in an external repository. Please rate, if applicable, the supporting data on the following criteria.

Is it accessible?

Yes

Is it clear?

Yes

Is it adequate?

Yes

Do you have any ethical concerns with this paper?

No

Comments to the Author

Comments to the authors

Helle et al. presents a study concerning the effect of the grandchildren's adverse life experiences (AELEs) on grandparental investment in modern humans. The study is based on the data collected from a survey with 1,566 English and Welsh adolescents aged 11-16. The authors investigated whether the grandparental investment was influenced by AELEs experienced by their grandchildren, and whether this investment differed according to four basic grandparental types. They found that the investment of paternal grandparents decreased with the increasing degree of adverse life events that their grandchildren experienced prior to the previous year. In accordance with the evolutionary theory, maternal grandmothers were less sensitive to increases in AELEs than other types of grandparents and their investment did not change as adverse life events of their grandchildren increased. Their theoretical view on grandparental investment is new and interesting. However, there are several issues in this study that should be clarified before its publishing.

Comments to address

First of all, the authors should clarify the definition of early life conditions. It is questionable whether the age of 11 to 16 years can be perceived as early life period. Henry and Ulijaszek (1996, Long-term Consequences of Early Environment) defined early development as the period from conception to developmental maturity of an individual. The earlier an individual is exposed to

adverse conditions, the stronger may be their effects (reviewed in Lindström 1999, *Trends Ecol. Evol.* 14: 343). Hence, I would rather disagree with the authors that the adolescent age of 11 to 16 years is the period of early life. In addition, the authors tested the potential effect of adverse life events on grandparental investment during a relatively short period of time, i.e. only during one year of the grandchild's life before the survey year. The observed differences between the investment of maternal and paternal grandparents and between the investment of maternal grandmothers and other grandparents seem to be relatively weak (see L. 293-300). The authors could find stronger effects and differences, if they would study adverse life events during a longer period than one year of grandchildren's life. Therefore, my overall feeling is that the title of the work does not reflect well the situation studied.

I agree with the authors that the balance between the fitness costs and benefits of grandparental investment is as important or even more important as the needs of grandoffspring. In addition, I do agree with them that the grandparents can reduce their investment when their grandoffspring challenge adverse life conditions. However, their results actually indicate that the maternal grandmothers compensated the negative effects of adverse life events in a grandchild's family (see Fig. 2B). Hence, both scenarios of grandparental investment (reduction and compensation) as adaptive responses to adverse life conditions in grandchildren are plausible in this case.

Similarly to other mammals, humans very likely optimize their resource allocation (either by increasing or decreasing investment in future generations) according to the actual conditions. For example in house mice that are as much reproductively adaptive as modern humans, this strategy is state-dependent and can even change during a relatively short period of time (see e.g. Dušek et al. 2017, *PLoS One* 12: e0173985). Therefore, I would recommend to modify the text of Introduction and Discussion sections, including both tested hypotheses (L. 135-140). The text of Introduction and Discussion (e.g. L. 342-344) could further benefit from discussing the effects of environmental and social conditions on grandparental investment in other mammals and primates in particular.

Some parts of the text, particularly in the sections Statistical analysis and Results, are quite difficult to understand. I would like to encourage the authors to simplify the text as much as possible. The authors should consider rewriting their text to highlight its biological meaning (e.g. L. 311-317). In addition to the tested predictors, grandparental investment could also be influenced by other so far neglected factors like socio-economic status of the grandmother, socio-economic status of the grandfather, socio-economic status of both parents, health status of the grandmother, health status of the grandfather, health status of both parents, mobility of the grandmother, mobility of the grandfather, grandparent's ethnicity and parent's ethnicity. From an evolutionary point of view, a very important factor likely is the certainty of parenthood. To address the potential influence of this specific factor, the authors could test the number of sequential mates of each parent that might indicate the level of their sexual promiscuity. This should be particularly important for the investment of paternal grandparents. All of these factors might significantly affect the results of their analyses. The authors should therefore at least discuss the potential effects of the factors listed above that were not mentioned in their text.

Minor comments

1. The authors should complete citations of the relevant literature in the following parts of the text: L. 88, 121-124, 337-339, 381.
2. Please explain the cited hypotheses thoroughly (L. 33, 110-111).
3. The authors should consider to reformulate and simplify the following sentence: "Finally, recent theoretical modelling of the evolution of early life effects on later performance has suggested that the social environment experienced by the individuals in childhood may be more important for their later performance than the abiotic environment [46], validating the importance of measuring the social aspects of the developmental environment while considering the context-specificity of the grandparental investment." (L. 96-101).
4. The aim of the study should be mentioned in the Introduction section, not in the Discussion section (L. 337-339). Please reformulate this text.
5. Please simplify the following sentence: "The novel contribution of the current research is that such patterns also seem to hold with respect to the AELEs of the grandchildren." (L. 373-375).

6. L. 399-401: I believe that this sentence belongs in the Results section.
7. L. 403-406: The following text needs a detailed explanation: "Had we assumed the confounder to exert positive or negative effects on both AELEs and grandparental investment, we would have observed increasingly negative estimates for the association between AELEs and the investment in grandchildren among the grandparents." Please consider its simplification.
8. Fig. 2: I recommend to modify the text of the Y axis legend. Please indicate the factor of the predicted marginal probability. Is it possible to complete data points in these two graphs?

Review form: Reviewer 3

Recommendation

Major revision is needed (please make suggestions in comments)

Scientific importance: Is the manuscript an original and important contribution to its field?

Acceptable

General interest: Is the paper of sufficient general interest?

Marginal

Quality of the paper: Is the overall quality of the paper suitable?

Good

Is the length of the paper justified?

Yes

Should the paper be seen by a specialist statistical reviewer?

No

Do you have any concerns about statistical analyses in this paper? If so, please specify them explicitly in your report.

No

It is a condition of publication that authors make their supporting data, code and materials available - either as supplementary material or hosted in an external repository. Please rate, if applicable, the supporting data on the following criteria.

Is it accessible?

Yes

Is it clear?

Yes

Is it adequate?

Yes

Do you have any ethical concerns with this paper?

No

Comments to the Author

There is an issue of causality to discuss here. It is very possible that adversity leads to lower investment by grandparents, particularly those that are "farther removed" (i.e. as shown in Table 2). However, it seems equally plausible that lower investment by grandparents may lead to more adversity - or at least, a less close-knit family network may be associated with a family that

experiences more adversity. One can imagine that families that are close-knit and where all grandparents are involved are relatively stable and don't experience adversity whereas the opposite is true where only the closet grandparents are involved.

I think that a better test would have been to assess how investment changed before and after adverse events. I'm sure this is nigh-on impossible to do retrospectively, but I think that only this sort of study could have truly got at the causal link between adversity and investment. The study doesn't show that "investment does/doesn't change with adversity", but that "investment is less in children with more adversity". This is a subtle difference, but it means that sentences like on line 344-345 "Paternal grandparents, however, clearly reduced their investment in grandchildren who had experienced more AELEs", are not accurate; instead, it should be something like ""Paternal grandparents invested less in grandchildren who had experienced more AELEs", to remove the insinuation that this is a reactionary change in investment to some event.

I wonder to what extent investment is dependent on the presence of each grandparent's partner? For example, one could imagine that grandfathers may be more likely to engage if their partner (i.e. the grandmother) is still alive, given the role that women tend to play in the social fabric of families? Then again, and straying into anecdote, if my mother were still alive there's no way that my father would come and visit me and my children as often as he does, so it could work both ways.

Similarly, one would expect that maternal grandmothers are the youngest grandparents, given mothers are on average younger than fathers. Does the fact that they may be the only remaining grandparent mean that they are likely to invest more time, whereas as other types of grandparents are less likely to be the only one left and hence are more likely to have to "compete" for time/investment?

How can the AELEs be separated from the conditions experienced by the grandparents themselves? For example, if a child experiences adverse events, a grandparent may invest less because they don't deem it worth it from a reproductive value point of view, but equally they may invest less because they themselves are experiencing adversity and so have fewer resources to invest. Bereavement, addiction and poverty could all impact on the grandparent as much as the child.

I'm having trouble understanding how the different contributions to the latent variable of grandparent investment were modelled. On lines 209-211 it's stated that the most relevant variable is financial investment, the loading of which is set to one. How strong is this relative to the other investment variables? How was the decision made that financial contributions were most important ("was regarded" does not really justify it)? How was the latent variable distributed?

Figure 2 is quite unhelpful. A cynic would say that it was there to make a result that was not quite supported by the analysis look as though it were statistically significant, and in any case the result is just as easily explained by the text and so the figure is largely redundant.

Further, it would be useful to know the distribution of the AELE score: the emphasis in the results is on comparisons between children with scores of 0 and 8, but are there actually many children at the extremes of the distribution?

Line 350: I think this is a bit of a push – the results trended in this direction, but I don't think the evidence is strong enough to make statements about the fitness benefits.

Decision letter (RSPB-2021-1798.R0)

27-Sep-2021

Dear Dr Helle:

I have now received three reviews of your manuscript RSPB-2021-1798 entitled "Matrilateral bias of grandparental investment in grandchildren persists despite the grandchildren's adverse early life experiences." Based on these reviews and the advice of the Associate Editor, as well as my read of your manuscript, I am rejecting it for publication in Proceedings B in its current form. I agree with the reviewers that this is an important topic, but there are a number of issues that the reviewers have identified. First, and to me the most important, is that your paper is written as if the link between AELEs and grandparental investment was causal, but your data set does not provide the data you would need to establish a causal link. In addition, as Reviewers 2 and 3 note, there are quite a few alternative explanations that are not ruled out by the data set and therefore must be considered. Finally, the manuscript, and in particular the statistical analysis, are extremely difficult to follow in places, so I encourage you to heavily revise your manuscript to improve clarity. We would be happy to see a reworked manuscript that answers these points, as well as each of the others raised by the reviewers. Please note that this is not a provisional acceptance and we will send your revision back out for review, with the original reviewers, should they be available.

Sincerely,
Dr Sarah Brosnan
Editor, Proceedings B
<mailto:proceedingsb@royalsociety.org>

Associate Editor
Board Member: 1
Comments to Author:
Dear authors,

The paper has been evaluated by three reviewers and there is quite a range of opinions.

I think the paper requires at the very least a major re-write. I am also concerned about the issue of causality as highlighted by Reviewer 3. Essentially, I fully agree with Reviewer 3's take on this, and I doubt it is possible to infer that paternal grandparents reduce investment in grandchildren with AELEs, it could be exactly the other way round. The findings can still be interesting but the whole outline of the paper should shift.

Reviewer(s)' Comments to Author:

Referee: 1

Comments to the Author(s)

Matrilateral bias of grandparental investment in grandchildren persists despite the grandchildren's adverse early life experiences.

This paper examines if grandparents invest more in grandchildren who have had adverse childhood experiences. They find that maternal grandmothers invest highly regardless of the adversity of the childhood while paternal grandparents reduce investment in adversely affected children. The authors use thoughtful statistical methods to get at the proposed causal effects. I was not familiar with the phantom confounder method but at face value it seems a sensible approach overall. It is well written and clear and I think this paper is good and a good fit for PRSB.

I have a few comments below:

Lines 49, 55, 387: Please replace 'Western' and 'modern-day' society with terms that depict the relevance of the context more precisely (e.g. low fertility, high income, post-industrial). Western has political connotations and modern-day is derogatory to lower income countries which are also 'modern-day' insofar as they are contemporary.

Lines 66-69 talk about the literature on grandparents influence on grandchildren's socioeconomic position as well as whether this is a function of grandchild needs. I was surprised that the recent review by Anderson et al 2018 was not referred to as it addresses both these topics extensively.

P11: Please provide proper justification for the control variables chosen. Most of them make sense to me (although not grandchild ethnicity or sex), nevertheless justification for each one should be clearly stated in the methods section.

Anderson, L, et al. 2018. 'Grandparent Effects in Educational Outcomes: A Systematic Review'. *Sociological Science* 5: 114-42.

Referee: 2

Comments to the Author(s)

Comments to the authors

Helle et al. presents a study concerning the effect of the grandchildren's adverse life experiences (AELEs) on grandparental investment in modern humans. The study is based on the data collected from a survey with 1,566 English and Welsh adolescents aged 11-16. The authors investigated whether the grandparental investment was influenced by AELEs experienced by their grandchildren, and whether this investment differed according to four basic grandparental types. They found that the investment of paternal grandparents decreased with the increasing degree of adverse life events that their grandchildren experienced prior to the previous year. In accordance with the evolutionary theory, maternal grandmothers were less sensitive to increases in AELEs than other types of grandparents and their investment did not change as adverse life events of their grandchildren increased. Their theoretical view on grandparental investment is

new and interesting. However, there are several issues in this study that should be clarified before its publishing.

Comments to address

First of all, the authors should clarify the definition of early life conditions. It is questionable whether the age of 11 to 16 years can be perceived as early life period. Henry and Ulijaszek (1996, Long-term Consequences of Early Environment) defined early development as the period from conception to developmental maturity of an individual. The earlier an individual is exposed to adverse conditions, the stronger may be their effects (reviewed in Lindström 1999, Trends Ecol. Evol. 14: 343). Hence, I would rather disagree with the authors that the adolescent age of 11 to 16 years is the period of early life. In addition, the authors tested the potential effect of adverse life events on grandparental investment during a relatively short period of time, i.e. only during one year of the grandchild's life before the survey year. The observed differences between the investment of maternal and paternal grandparents and between the investment of maternal grandmothers and other grandparents seem to be relatively weak (see L. 293-300). The authors could find stronger effects and differences, if they would study adverse life events during a longer period than one year of grandchildren's life. Therefore, my overall feeling is that the title of the work does not reflect well the situation studied.

I agree with the authors that the balance between the fitness costs and benefits of grandparental investment is as important or even more important as the needs of grandoffspring. In addition, I do agree with them that the grandparents can reduce their investment when their grandoffspring challenge adverse life conditions. However, their results actually indicate that the maternal grandmothers compensated the negative effects of adverse life events in a grandchild's family (see Fig. 2B). Hence, both scenarios of grandparental investment (reduction and compensation) as adaptive responses to adverse life conditions in grandchildren are plausible in this case.

Similarly to other mammals, humans very likely optimize their resource allocation (either by increasing or decreasing investment in future generations) according to the actual conditions. For example in house mice that are as much reproductively adaptive as modern humans, this strategy is state-dependent and can even change during a relatively short period of time (see e.g. Dušek et al. 2017, PLoS One 12: e0173985). Therefore, I would recommend to modify the text of Introduction and Discussion sections, including both tested hypotheses (L. 135-140). The text of Introduction and Discussion (e.g. L. 342-344) could further benefit from discussing the effects of environmental and social conditions on grandparental investment in other mammals and primates in particular.

Some parts of the text, particularly in the sections Statistical analysis and Results, are quite difficult to understand. I would like to encourage the authors to simplify the text as much as possible. The authors should consider rewriting their text to highlight its biological meaning (e.g. L. 311-317). In addition to the tested predictors, grandparental investment could also be influenced by other so far neglected factors like socio-economic status of the grandmother, socio-economic status of the grandfather, socio-economic status of both parents, health status of the grandmother, health status of the grandfather, health status of both parents, mobility of the grandmother, mobility of the grandfather, grandparent's ethnicity and parent's ethnicity. From an evolutionary point of view, a very important factor likely is the certainty of parenthood. To address the potential influence of this specific factor, the authors could test the number of sequential mates of each parent that might indicate the level of their sexual promiscuity. This should be particularly important for the investment of paternal grandparents. All of these factors might significantly affect the results of their analyses. The authors should therefore at least discuss the potential effects of the factors listed above that were not mentioned in their text.

Minor comments

1. The authors should complete citations of the relevant literature in the following parts of the text: L. 88, 121-124, 337-339, 381.
2. Please explain the cited hypotheses thoroughly (L. 33, 110-111).
3. The authors should consider to reformulate and simplify the following sentence: "Finally, recent theoretical modelling of the evolution of early life effects on later performance has suggested that the social environment experienced by the individuals in childhood may be more

important for their later performance than the abiotic environment [46], validating the importance of measuring the social aspects of the developmental environment while considering the context-specificity of the grandparental investment." (L. 96-101).

4. The aim of the study should be mentioned in the Introduction section, not in the Discussion section (L. 337-339). Please reformulate this text.

5. Please simplify the following sentence: "The novel contribution of the current research is that such patterns also seem to hold with respect to the AELEs of the grandchildren." (L. 373-375).

6. L. 399-401: I believe that this sentence belongs in the Results section.

7. L. 403-406: The following text needs a detailed explanation: "Had we assumed the confounder to exert positive or negative effects on both AELEs and grandparental investment, we would have observed increasingly negative estimates for the association between AELEs and the investment in grandchildren among the grandparents." Please consider its simplification.

8. Fig. 2: I recommend to modify the text of the Y axis legend. Please indicate the factor of the predicted marginal probability. Is it possible to complete data points in these two graphs?

Referee: 3

Comments to the Author(s)

There is an issue of causality to discuss here. It is very possible that adversity leads to lower investment by grandparents, particularly those that are "farther removed" (i.e. as shown in Table 2). However, it seems equally plausible that lower investment by grandparents may lead to more adversity - or at least, a less close-knit family network may be associated with a family that experiences more adversity. One can imagine that families that are close-knit and where all grandparents are involved are relatively stable and don't experience adversity whereas the opposite is true where only the closet grandparents are involved.

I think that a better test would have been to assess how investment changed before and after adverse events. I'm sure this is nigh-on impossible to do retrospectively, but I think that only this sort of study could have truly got at the causal link between adversity and investment. The study doesn't show that "investment does/doesn't change with adversity", but that "investment is less in children with more adversity". This is a subtle difference, but it means that sentences like on line 344-345 "Paternal grandparents, however, clearly reduced their investment in grandchildren who had experienced more AELEs", are not accurate; instead, it should be something like "Paternal grandparents invested less in grandchildren who had experienced more AELEs", to remove the insinuation that this is a reactionary change in investment to some event.

I wonder to what extent investment is dependent on the presence of each grandparent's partner? For example, one could imagine that grandfathers may be more likely to engage if their partner (i.e. the grandmother) is still alive, given the role that women tend to play in the social fabric of families? Then again, and straying into anecdote, if my mother were still alive there's no way that my father would come and visit me and my children as often as he does, so it could work both ways.

Similarly, one would expect that maternal grandmothers are the youngest grandparents, given mothers are on average younger than fathers. Does the fact that they may be the only remaining grandparent mean that they are likely to invest more time, whereas as other types of grandparents are less likely to be the only one left and hence are more likely to have to "compete" for time/investment?

How can the AELEs be separated from the conditions experienced by the grandparents themselves? For example, if a child experiences adverse events, a grandparent may invest less because they don't deem it worth it from a reproductive value point of view, but equally they may invest less because they themselves are experiencing adversity and so have fewer resources to invest. Bereavement, addiction and poverty could all impact on the grandparent as much as the child.

I'm having trouble understanding how the different contributions to the latent variable of grandparent investment were modelled. On lines 209-211 it's stated that the most relevant variable is financial investment, the loading of which is set to one. How strong is this relative to the other investment variables? How was the decision made that financial contributions were most important ("was regarded" does not really justify it)? How was the latent variable distributed?

Figure 2 is quite unhelpful. A cynic would say that it was there to make a result that was not quite supported by the analysis look as though it were statistically significant, and in any case the result is just as easily explained by the text and so the figure is largely redundant.

Further, it would be useful to know the distribution of the AELE score: the emphasis in the results is on comparisons between children with scores of 0 and 8, but are there actually many children at the extremes of the distribution?

Line 350: I think this is a bit of a push – the results trended in this direction, but I don't think the evidence is strong enough to make statements about the fitness benefits.

Author's Response to Decision Letter for (RSPB-2021-1798.R0)

See Appendix A.

RSPB-2021-2574.R0

Review form: Reviewer 3

Recommendation

Accept with minor revision (please list in comments)

Scientific importance: Is the manuscript an original and important contribution to its field?

Good

General interest: Is the paper of sufficient general interest?

Acceptable

Quality of the paper: Is the overall quality of the paper suitable?

Good

Is the length of the paper justified?

Yes

Should the paper be seen by a specialist statistical reviewer?

No

Do you have any concerns about statistical analyses in this paper? If so, please specify them explicitly in your report.

No

It is a condition of publication that authors make their supporting data, code and materials available - either as supplementary material or hosted in an external repository. Please rate, if applicable, the supporting data on the following criteria.

Is it accessible?

Yes

Is it clear?

Yes

Is it adequate?

Yes

Do you have any ethical concerns with this paper?

No

Comments to the Author

The revised manuscript is an interesting read and deals with most of the comments I made on the original version. One important issue remains, however.

I'm afraid I have to return to the issue of causality here, which I raised in my previous review – i.e. the current interpretation is that adversity \square lower investment, but one could also argue that lower investment \square adversity. I fully accept that the data are not there to disentangle these effects and that, as the response letter states, it comes down to which is more plausible. The response letter also states that since, in the analysis, adverse events were measured before investment, it make sense to assume that adversity \square investment. One might imagine that investment in two consecutive years is correlated, as is adversity in two consecutive years. If the analysis were repeated with last year's investment and this year's adversity and the same results were gained, it would suggest that causality is hard to infer; if this analysis revealed different results, it would suggest that that the assertions about causality are correct. I am not imagining for one minute that this analysis should be repeated in this way, but I would like to see this issue tackled head-on in the manuscript rather than somewhat brushed off as it appears to have been in the original response letter. This needs to go in the "limitations" paragraph currently on lines 412-423.

Decision letter (RSPB-2021-2574.R0)

13-Jan-2022

Dear Dr Helle

I am pleased to inform you that your manuscript RSPB-2021-2574 entitled "Matrilateral bias of grandparental investment in grandchildren persists despite the grandchildren's adverse early life experiences" has been accepted for publication in Proceedings B pending some minor revisions. I am in general very pleased with your revision. However, I agree with Reviewer 1 that you should still acknowledge the fact that it is difficult to be sure about causality with this type of analyses and that you should perhaps address this directly as a potential limitation of the study. I strongly encourage you to follow this suggestion. Overall, however, I think this study presents interesting results that are likely to advance the field. Please respond to the referee(s)' comments and revise your manuscript accordingly. Because the schedule for publication is very tight, it is a condition of publication that you submit the revised version of your manuscript within 7 days. If you do not think you will be able to meet this date please let us know.

To revise your manuscript, log into <https://mc.manuscriptcentral.com/prsb> and enter your Author Centre, where you will find your manuscript title listed under "Manuscripts with

Decisions." Under "Actions," click on "Create a Revision." Your manuscript number has been appended to denote a revision. You will be unable to make your revisions on the originally submitted version of the manuscript. Instead, revise your manuscript and upload a new version through your Author Centre.

If you wish to submit your data to Dryad (<http://datadryad.org/>) and have not already done so you can submit your data via this link [http://datadryad.org/submit?journalID=RSPB&manu=\(Document not available\)](http://datadryad.org/submit?journalID=RSPB&manu=(Document%20not%20available)) which will take you to your unique entry in the Dryad repository. If you have already submitted your data to dryad you can make any necessary revisions to your dataset by following the above link. Please see <https://royalsociety.org/journals/ethics-policies/data-sharing-mining/> for more details.

Sincerely,
Dr Sarah Brosnan
Editor, Proceedings B
mailto:proceedingsb@royalsociety.org

Reviewer(s)' Comments to Author:

Referee: 3

Comments to the Author(s).

The revised manuscript is an interesting read and deals with most of the comments I made on the original version. One important issue remains, however.

I'm afraid I have to return to the issue of causality here, which I raised in my previous review – i.e. the current interpretation is that adversity \square lower investment, but one could also argue that lower investment \square adversity. I fully accept that the data are not there to disentangle these effects and that, as the response letter states, it comes down to which is more plausible. The response letter also states that since, in the analysis, adverse events were measured before investment, it make sense to assume that adversity \square investment. One might imagine that investment in two consecutive years is correlated, as is adversity in two consecutive years. If the analysis were repeated with last year's investment and this year's adversity and the same results were gained, it would suggest that causality is hard to infer; if this analysis revealed different results, it would suggest that the assertions about causality are correct. I am not imagining for one minute that this analysis should be repeated in this way, but I would like to see this issue tackled head-on in the manuscript rather than somewhat brushed off as it appears to have been in the original response letter. This needs to go in the "limitations" paragraph currently on lines 412-423.

Author's Response to Decision Letter for (RSPB-2021-2574.R0)

See Appendix B.

Decision letter (RSPB-2021-2574.R1)

18-Jan-2022

Dear Dr Helle

I am pleased to inform you that your manuscript entitled "Matrilateral bias of grandparental investment in grandchildren persists despite the grandchildren's adverse early life experiences" has been accepted for publication in Proceedings B.

Data Accessibility section

Open Access

Paper charges

Sincerely,

Proceedings B

Appendix A

Samuli Helle, PhD

Department of Social Research
University of Turku
FIN-20014 Turku, Finland
Tel. +358 2 333 6559
email. sayrhe@utu.fi

November 26, 2021

Dear Prof. Brosnan

Thank you very much for the opportunity to revise our manuscript “Matrilateral bias of grandparental investment in grandchildren persists despite the grandchildren’s adverse early life experiences.” (RSPB-2021-1798). Please, below find our specific responses to helpful comments by the editorial board member and three reviewers that have strengthened our manuscript. We hope that you now find the manuscript suitable for publication in Proceedings B.

Responses to editorial board member:

The paper has been evaluated by three reviewers and there is quite a range of opinions.

I think the paper requires at the very least a major re-write. I am also concerned about the issue of causality as highlighted by Reviewer 3. Essentially, I fully agree with Reviewer 3's take on this, and I doubt it is possible to infer that paternal grandparents reduce investment in grandchildren with AELEs, it could be exactly the other way round. The findings can still be interesting but the whole outline of the paper should shift.

Response: In our revision, we have made our best to tackle all these points.

Responses to reviewer #1

This paper examines if grandparents invest more in grandchildren who have had adverse childhood experiences. They find that maternal grandmothers invest highly regardless of the adversity of the childhood while paternal grandparents reduce investment in adversely affected children. They authors use thoughtful statistical methods to get at the proposed causal effects. I was not familiar with the phantom confounder method but at face value it seems a sensible approach overall. It is well written and clear and I think this paper is good and a good fit for PRSB.

Response: Thank you. We would like to emphasize that we do not claim to report true causal effects here, like stated in the lines 406-410: “However, we do not claim to have revealed causal effects here, as all the effects modelled were linear and causal inference is inherently non-parametric in nature [81].” But we have made every effort to outrule the well-known threat to causal inference in non-experimental data, namely excluded confounders and measurement error in independent variables. In the revised ms, we have tried to be more explicit about this along the lines suggested by the editorial board member and reviewer #3.

I have a few comments below:

Lines 49, 55, 387: Please replace ‘Western’ and ‘modern-day’ society with terms that depict the relevance of the context more precisely (e.g. low fertility, high income, post-industrial). Western has political connotations and modern-day is derogatory to lower income countries which are also ‘modern-day’ insofar as they are contemporary.

Response: Thank you for that suggestion, we have replaced these words.

Lines 66-69 talk about the literature on grandparents influence on grandchildren's socioeconomic position as well as whether this is a function of grandchild needs. I was surprised that the recent review by Anderson et al 2018 was not referred to as it addresses both these topics extensively.

Response: We thank the reviewer for bringing this article into our attention. It has now been cited in the main text (please see line 70 and ref #25).

P11: Please provide proper justification for the control variables chosen. Most of them make sense to me (although not grandchild ethnicity or sex), nevertheless justification for each one should be clearly stated in the methods section.

Response: The variables mentioned by the reviewer are not considered as statistical controls, or causal confounders, but as precision covariates (or competing treatments) aimed to reduce the error variance in grandparental investment in grandchildren (see lines 222-241). The rationale for their inclusion is thus not handling confounding but just to reduce the uncertainty of the regression coefficients of main interest here. Since in our opinion the current data sets do not include potential causal confounders of the effect of AELEs on grandparental investment except for the age of grandchildren (see lines 232-236), we have used the sensitivity analysis to model the effect of potential unmeasured confounders and concluded that their estimated effects likely play no important role here.

Responses to reviewer #2

Helle et al. presents a study concerning the effect of the grandchildren’s adverse life experiences (AELEs) on grandparental investment in modern humans. The study is based on the data collected from a survey with 1,566 English and Welsh adolescents aged 11-16. The authors investigated whether the grandparental investment was influenced by AELEs experienced by their grandchildren, and whether this investment differed according to four basic grandparental types. They found that the investment of paternal grandparents decreased with the increasing degree of adverse life events that their grandchildren experienced prior to the previous year. In accordance with the evolutionary theory, maternal grandmothers were less sensitive to increases in AELEs than other types of grandparents and their investment did not change as adverse life events of their grandchildren increased. Their theoretical view on grandparental investment is new and interesting. However, there are several issues in this study that should be clarified before its publishing.

Response: Thank you, we are glad to hear you feel our perspective is new, interesting and worth pursuing.

Comments to address

First of all, the authors should clarify the definition of early life conditions. It is questionable whether the age of 11 to 16 years can be perceived as early life period. Henry and Ulijaszek (1996, Long-term Consequences of Early Environment) defined early development as the period from conception to developmental maturity of an individual. The earlier an individual is exposed to adverse conditions, the stronger may be their effects (reviewed in Lindström 1999, Trends Ecol. Evol. 14: 343). Hence, I would rather disagree with the authors that the adolescent age of 11 to 16 years is the period of early life. In addition, the authors tested the potential effect of adverse life events on grandparental investment during a relatively short period of time, i.e. only during one year of the grandchild's life before the survey year. The observed differences between the investment of maternal and paternal grandparents and between the investment of maternal grandmothers and other grandparents seem to be relatively weak (see L. 293-300). The authors could find stronger effects and differences, if they would study adverse life events during a longer period than one year of grandchildren's life. Therefore, my overall feeling is that the title of the work does not reflect well the situation studied.

Response: We have better defined the meaning of early life conditions in the revised ms (please see lines 121, 143 and 167-170). In this study, that period covers the already lived life of the children. So the reviewer has misunderstood this point by thinking that the adverse early life events were recorded only for the period of 11 to 16 years of the children. We have clarified that the adolescents and children in this study ranged from 11 to 16 years of age (line 143). Based on the current data, we unfortunately have no way of timing those events with respect to the age of children.

Unfortunately, we can't confirm or refute the reviewer's comment that "*the authors tested the potential effect of adverse life events on grandparental investment during a relatively short period of time, i.e. only during one year of the grandchild's life before the survey year*" because do not know how the children really understood the questions like "how often you see them?". The original collectors of the data do not explicitly state what time periods the children *should* consider when answering those investment questions. We think the reviewer is correct that the investments considered here do most likely reflect the grandchildren's experiences of the near past but we can't say with full confidence that they would be restricted to one year before the survey year. On the other hand, the fact that AELEs recorded took place before the expected evaluation of grandparental investment is relevant for the interpretation of our results as potential causal estimates (please see our response to the reviewer #3's comments below).

I agree with the authors that the balance between the fitness costs and benefits of grandparental investment is as important or even more important as the needs of grandoffspring. In addition, I do agree with them that the grandparents can reduce their investment when their grandoffspring challenge adverse life conditions. However, their results actually indicate that the maternal grandmothers compensated the negative effects of adverse life events in a grandchild's family (see Fig. 2B). Hence, both scenarios of grandparental investment (reduction and compensation) as adaptive responses to adverse live conditions in grandchildren are plausible in this case. Similarly to other mammals, humans very likely optimize their resource allocation (either by increasing or decreasing investment

in future generations) according to the actual conditions. For example in house mice that are as much reproductively adaptive as modern humans, this strategy is state-dependent and can even change during a relatively short period of time (see e.g. Dušek et al. 2017, PLoS One 12: e0173985). Therefore, I would recommend to modify the text of Introduction and Discussion sections, including both tested hypotheses (L. 135-140). The text of Introduction and Discussion (e.g. L. 342-344) could further benefit from discussing the effects of environmental and social conditions on grandparental investment in other mammals and primates in particular.

Response: We are glad to see that the reviewer supports our investigation of life history theory and grandparental investment perspectives together. However, although the idea of compensation is interesting, we do not fully agree with this interpretation of our results. The reason is that based on our understanding of how compensation works, we should have found maternal grandmothers to increase their investment and not to decrease it as seen in Table 2 (and in former Figure 2B, which has been deleted from the revised version of the ms). Therefore, we see the compensation explanation as akin to the “need” hypothesis, for which we don’t find support from these data. Because of this, we have not incorporated the idea of compensation into introduction as a separate concept but as a part of the “need” hypothesis (see line 69). We have also added two more references into this paragraph as examples of the “need/compensation” concept (i.e., Sheppard & Sear 2016 and Barnett et al. 2010). If we have understood the concept of compensation the reviewer had in mind wrongly, we are happy to revise our text.

Sheppard P, Sear R. 2016 Do grandparents compete with or support their grandchildren? In Guatemala, paternal grandmothers may compete, and maternal grandmothers may cooperate. *R. Soc. open sci.* 3, 160069. (doi: 10.1098/rsos.160069)

Barnett MA, Scaramella LV, Neppl TK, Ontai LL, Conger RD. 2010 Grandmother Involvement as a Protective Factor for Early Childhood Social Adjustment. *J. Fam. Psychol.* 24, 635-645. (doi: 10.1037/a0020829)

We have also added one reference in the discussion that, to our knowledge, is the only non-human example of how the influence of post-reproductive females could be regarded extra valuable in poor resource settings (please see lines 380-382). If we have missed some relevant references regarding this topic, we are happy to include more references.

Brent LNJ, Franks DW, Foster EA, Balcomb KC, Cant MA, Croft D. 2015 Ecological knowledge, leadership, and the evolution of menopause in killer whales. *Curr. Biol.* 25, 746-750. (doi: 10.1016/j.cub.2015.01.037)

Some parts of the text, particularly in the sections Statistical analysis and Results, are quite difficult to understand. I would like to encourage the authors to simplify the text as much as possible. The authors should consider rewriting their text to highlight its biological meaning (e.g. L. 311-317). In addition to the tested predictors, grandparental investment could also be influenced by other so far neglected factors like socio-economic status of the grandmother, socio-economic status of the grandfather, socio-economic status of both parents, health status of the grandmother, health status of the grandfather, health status of both parents, mobility of the grandmother, mobility of the grandfather, grandparent’s ethnicity and parent’s ethnicity. From an evolutionary point of view, a very important factor likely is the certainty

of parenthood. To address the potential influence of this specific factor, the authors could test the number of sequential mates of each parent that might indicate the level of their sexual promiscuity. This should be particularly important for the investment of paternal grandparents. All of these factors might significantly affect the results of their analyses. The authors should therefore at least discuss the potential effects of the factors listed above that were not mentioned in their text.

Response: Thank you for the opportunity to clarify further. We have removed some of the more technical parts into the electronic supplementary materials (e.g. figure 1 and information related to it). Also, the paragraph mentioned by the reviewer has been modified for clarity (see lines 313-318). Finally, the reviewer is correct by saying that several factors not recorded in the current data (and in most other data) may play a role in the associations studied here. We were aware of this issue from the beginning of our analysis. For this reason we performed the sensitivity analysis to evaluate the importance of omitted variable bias. There was no obvious evidence that this was the case here. We thus see no need to discuss each of these factors in the discussion as they are usefully treated as a group and the article is already very long. Moreover, the factors mentioned by the reviewer seem to rather be covariates influencing grandparental investment than real statistical confounders.

The paternity uncertainty likely plays some role here (and certainly in evolutionary terms) and our findings are in agreement with the predictions from paternity uncertainty as maternal grandmothers have the highest level of paternity certainty and they showed the highest level of investment even in the case of several adverse early life events. The paternity uncertainty is built into the predicted patterning of grandparental investment across grandparent types described in this manuscript. However, the request made by the reviewer with respect to the number of sequential mates cannot be done using the current data.

Minor comments

1. The authors should complete citations of the relevant literature in the following parts of the text: L. 88, 121-124, 337-339, 381.

Response: We think the ref #33 is completely appropriate here (i.e., exactly what we mean and a significant reference). With respect to lines 121-124 and 337-339, we have added the key reference (see lines 124 and 340).

2. Please explain the cited hypotheses thoroughly (L. 33, 110-111).

Response: We are afraid that we have no space to elaborate the hypotheses further in the abstract. Moreover, we do not explain e.g. sex-chromosomal selection hypothesis in more detail here because the hypotheses tested here do not consider grandchild sex.

3. The authors should consider to reformulate and simplify the following sentence: “Finally, recent theoretical modelling of the evolution of early life effects on later performance has suggested that the social environment experienced by the individuals in childhood may be more important for their later performance than the abiotic environment [46], validating the importance of measuring the social aspects of the developmental environment while considering the context-specificity of the grandparental investment.” (L. 96-101).

Response: We have re-written the sentence as: “Finally, recent theoretical modelling of the evolution of early life effects on later performance has suggested that the social environment experienced by the individuals in childhood may be more important for their later performance than the abiotic environment [46]. This finding emphasises the importance of measuring the social aspects of the developmental environment while considering the context-specificity of the grandparental investment.”

4. The aim of the study should be mentioned in the Introduction section, not in the Discussion section (L. 337-339). Please reformulate this text.

Response: We do this also in the introduction, please see the paragraph starting from the line 119.

5. Please simplify the following sentence: “The novel contribution of the current research is that such patterns also seem to hold with respect to the AELEs of the grandchildren.” (L. 373-375).

Response: This sentence has been rewritten as: “The novel contribution of the current research is that for grandchildren who experienced many AELEs the importance of maternal grandparents and especially maternal grandmothers still seem to hold.” (see lines 372-374).

6. L. 399-401: I believe that this sentence belongs in the Results section.

Response: We think it is a good idea to repeat here what we did in the results.

7. L. 403-406: The following text needs a detailed explanation: “Had we assumed the confounder to exert positive or negative effects on both AELEs and grandparental investment, we would have observed increasingly negative estimates for the association between AELEs and the investment in grandchildren among the grandparents.”. Please consider its simplification.

Response: This sentence has been deleted for clarity.

8. Fig. 2: I recommend to modify the text of the Y axis legend. Please indicate the factor of the predicted marginal probability. Is it possible to complete data points in these two graphs?

Response: Based on the suggestion made by the reviewer #3 (which we agreed on), we have deleted the figure 2.

Responses to reviewer #3

There is an issue of causality to discuss here. It is very possible that adversity leads to lower investment by grandparents, particularly those that are “farther removed” (i.e. as shown in Table 2). However, it seems equally plausible that lower investment by grandparents may lead to more adversity – or at least, a less close-knit family network may be associated with a family that experiences more adversity. One can imagine that families that are close-knit and where all grandparents are involved are relatively stable and don’t experience adversity whereas the opposite is true where only the closest grandparents are involved.

Response: This is true. Since the direction of causality cannot be settled with the current approach (or in SEM in general without instrumental variables, which we do not have), such arguments come down to which direction is more plausible. Our rationale for looking at how AELEs affect grandparental investment (and not vice versa) is based on the logic how these variables were likely defined in our data. The questions on grandparental investment were largely framed to measure current investment of grandparents (e.g. “how often you see them” and not “how often have you seen them in previous x years” etc), whereas the adverse early life events that were considered here are events that took place before the last year (i.e., “have you experienced any these before last year?”). So the temporal ordering of independent and dependent variables in our analysis satisfies the basic assumption needed for causal inference: the cause(s) before the effect(s). We do agree that close-knit families probably are more stable, however, it is equally the case that adversity may lead to less close knit families due to the impact/nature of the stressors involved. But as just described the temporal ordering of the variables recorded in the current study do not allow us to analyse such setups.

I think that a better test would have been to assess how investment changed before and after adverse events. I’m sure this is nigh-on impossible to do retrospectively, but I think that only this sort of study could have truly got at the causal link between adversity and investment. The study doesn’t show that “investment does/doesn’t change with adversity”, but that “investment is less in children with more adversity”. This is a subtle difference, but it means that sentences like on line 344-345 “Paternal grandparents, however, clearly reduced their investment in grandchildren who had experienced more AELEs”, are not accurate; instead, it should be something like “Paternal grandparents invested less in grandchildren who had experienced more AELEs”, to remove the insinuation that this is a reactionary change in investment to some event.

Response: This is a good point thank you, we fully agree. Our current analysis is totally constrained by the available data that has not been collected with a causal analysis in mind (and not by the current authors). Therefore, we have tried our best to exclude the well-known threats to causal inference in cross-sectional correlative data. We do not anywhere claim to have revealed causal effects here (which are non-parametric by definition and our analyses deal with linear associations only). Consistent with Reviewer’s suggestion, we have changed our wording throughout the discussion to make this point clearer (please see e.g. lines 341-344).

I wonder to what extent investment is dependent on the presence of each grandparent’s partner? For example, one could imagine that grandfathers may be more likely to engage if their partner (i.e. the grandmother) is still alive, given the role that women tend to play in the social fabric of families? Then again, and straying into anecdote, if my mother were still alive there’s no way that my father would come and visit me and my children as often as he does, so it could work both ways.

Response: We are currently working on another publication where we look at such associations. Based on those results, we do not see any indication that e.g. grandfather’s investment in grandchildren would be increased in the case of a living grandmother, i.e., they seem not to be “incidentally exposed” as suggested in the social scientific literature. Not at least in the current sample.

Similarly, one would expect that maternal grandmothers are the youngest grandparents, given mothers are on average younger than fathers. Does the fact that they may be the only remaining grandparent mean that they are likely to invest more time, whereas as other types of grandparents are less likely to be the only one left and hence are more likely to have to “compete” for time/investment?

Response: Our model controls for grandparental age so we do not think this is an issue here.

How can the AELEs be separated from the conditions experienced by the grandparents themselves? For example, if a child experiences adverse events, a grandparent may invest less because they don't deem it worth it from a reproductive value point of view, but equally they may invest less because they themselves are experiencing adversity and so have fewer resources to invest. Bereavement, addiction and poverty could all impact on the grandparent as much as the child.

Response: This relates to our earlier point that AELEs for the grandchild are likely to have intergenerational concomitants. As shown on lines 167-181, most of the questions asked clearly concerned the grandchild itself or his/her parents. There are two questions (i.e., “someone in the family died” and “family had drug/alcohol problem”) where the reference is made for the “family”, in which case it is more unclear how the children defined the meaning of “family”, i.e. did they include grandparent to their family. So in general we think that the grandparental generation is not commonly considered with respect to AELEs but naturally we cannot be totally sure about this and we have no data to examine this question further. Moreover, our sensitivity analysis described in lines 268-291 deals with such a scenario and showed that such a scenario is unlikely to affect our conclusions.

I'm having trouble understanding how the different contributions to the latent variable of grandparent investment were modelled. On lines 209-211 it's stated that the most relevant variable is financial investment, the loading of which is set to one. How strong is this relative to the other investment variables? How was the decision made that financial contributions were most important (“was regarded” does not really justify it)? How was the latent variable distributed?

Response: The loading of financial investment was fixed to unity to set the scale to the latent variable because we thought that, compared to the other variables available reflecting grandparental investment, financial investment is most easily understood as investment (with respect to all grandparental types). So our justification here is fully theoretical. It is important to acknowledge that this does not mean that the actual “meaning” of the latent variable is dependent on this selection, just its scale is and we need to fix some indicator's loading (or factor variance) to identify the latent variable. As said in the methods (lines 210-213): “...loadings can be interpreted as the extent to which a one-unit increase in the latent variable score (changes the predicted probit index in standard deviation units...” and thus those loadings are readily comparable. The (continuous) latent variable is assumed to have a Gaussian (normal) distribution.

Figure 2 is quite unhelpful. A cynic would say that it was there to make a result that was not quite supported by the analysis look as though it were statistically significant, and in any case the result is just as easily explained by the text and so the figure is largely redundant.

Response: We agree and have thus deleted the Figure 2.

Further, it would be useful to know the distribution of the AELE score: the emphasis in the results is on comparisons between children with scores of 0 and 8, but are there actually many children at the extremes of the distribution?

Response: No, there were very few children who scored 8 for AELEs and much more children with zero AELEs (please see the histogram below). It should be acknowledged that our purpose was not to emphasize this range but just to show the effect at the extremes of the range of AELEs. The analysis performed here, like other regression model, make no distributional assumptions on independent variables.

Line 350: I think this is a bit of a push – the results trended in this direction, but I don't think the evidence is strong enough to make statements about the fitness benefits.

Response: Our intention was not to imply any fitness-benefits, based on the current findings. The sentence has been softened: "...should relate to their potential costs and benefits, as these eventually could determine the payoff in terms of their grandchildren's evolutionary fitness" (see lines 351-353).

Yours faithfully,

Dr. Samuli Helle
On the behalf of other co-authors

Appendix B

Samuli Helle, PhD

Department of Social Research
University of Turku
FIN-20014 Turku, Finland
Tel. +358 2 333 6559
email. sayrhe@utu.fi

January 17, 2022

Dear Prof. Brosnan

Thank you very much for accepting our manuscript “Matrilateral bias of grandparental investment in grandchildren persists despite the grandchildren’s adverse early life experiences” (RSPB-2021-2574) pending minor revision. Please, below find our responses to your comments and to the final comment made by the reviewer. We hope that you now find the manuscript suitable for publication in Proceedings B.

Responses to editorial board member:

I am in general very pleased with your revision. However, I agree with Reviewer 1 that you should still acknowledge the fact that it is difficult to be sure about causality with this type of analyses and that you should perhaps address this directly as a potential limitation of the study. I strongly encourage you to follow this suggestion. Overall, however, I think this study presents interesting results that are likely to advance the field. Please respond to the referee(s)' comments and revise your manuscript accordingly.

Response: Thank you. As requested, we have added this point into the paragraph where we discuss the limitations of this study and data (see the lines 418-425).

Responses to reviewer #3

The revised manuscript is an interesting read and deals with most of the comments I made on the original version. One important issue remains, however.

I’m afraid I have to return to the issue of causality here, which I raised in my previous review – i.e. the current interpretation is that adversity \diamond lower investment, but one could also argue that lower investment \diamond adversity. I fully accept that the data are not there to disentangle these effects and that, as the response letter states, it comes down to which is more plausible. The response letter also states that since, in the analysis, adverse events were measured before investment, it make sense to assume that adversity \diamond investment. One might imagine that investment in two consecutive years is correlated, as is adversity in two consecutive years. If the analysis were repeated with last year’s investment and this year’s adversity and the same results were gained, it would suggest that causality is hard to infer; if this analysis revealed different results, it would suggest that that the assertions about causality are correct. I am not imagining for one minute that this analysis should be repeated in this way, but I would like to see this issue tackled head-on in the manuscript rather than somewhat brushed

off as it appears to have been in the original response letter. This needs to go in the “limitations” paragraph currently on lines 412-423.

Response: Thank you. We have now emphasized the points suggested in the limitations paragraph (see especially the lines 418-425).

Yours faithfully,

Dr. Samuli Helle
On the behalf of other co-authors